# Characterization of the diversity of barn owl's mitochondrial genome reveals high copy number variations in the control region

**Marcelo J. Serrano[1], Jérôme Goudet[1,2], Tristan Cumer [1,2]***

**1** Department of Ecology and Evolution, University of Lausanne, Lausanne, Switzerland, **2** Swiss Institute of Bioinformatics, Lausanne, Switzerland

\* t.cumer.sci@gmail.com

**Data Availability Statement:** All sequencing files are available from the GenBank database (BioProjects: PRJNA727977; PRJNA700797;

## Abstract

Mitochondria are known to play an essential role in the cell. These organelles contain their own DNA, which is divided in a coding and non-coding region (NCR). While much of the NCR's function is unknown, tandem repeats have been observed in several vertebrates, with extreme intra-individual, intraspecific and interspecific variation. Taking advantage of a new complete reference for the mitochondrial genome of the Afro-European Barn Owl (*Tyto alba*), as well as 172 whole genome-resequencing; we (i) describe the reference mitochondrial genome with a special focus on the repeats in the NCR, (ii) quantify the variation in number of copies between individuals, and (iii) explore the possible factors associated with the variation in the number of repetitions. The reference mitochondrial genome revealed a *long* (256bp) and a *short* (80bp) tandem repeat in the NCR region. The re-sequenced genomes showed a great variation in number of copies between individuals, with 4 to 38 copies of the *Long* and 6 to 135 copies of the *short* repeat. Among the factors associated with this variation between individuals, the tissue used for extraction was the most significant. The exact mechanisms of the formations of these repeats are still to be discovered and understanding them will help explain the maintenance of the polymorphism in the number of copies, as well as their interactions with the metabolism, the aging and health of the individuals.

## Introduction

Mitochondria are cytoplasmic organelles that have been extensively studied due to their essential role in the cell [1]. These functions include the oxidative phosphorylation activity, responsible for the generation of the energetic intermediates adenosine triphosphate (ATP) [2]. The enzymes involved in the oxidative phosphorylation are encoded by genes present in the mitochondrial DNA (mtDNA). This mitochondrial DNA is a double stranded circular molecule with a mutation rate higher than nuclear DNA, either due to the lack of histones, the near absence of DNA repair systems and/or its exposure to oxygen radicals during oxidative phosphorylation activity [3]. This genetic material is characterized by a coding and a Non-Coding

PRJNA727915; PRJNA925445). All other relevant data are within the paper, its Supporting Information files as well as online at https://github.com/cumtr/Serrano_et_al_2023_Diversity_mitochondrial_genome_Barn_Owl.

**Funding:** This work was supported by the Swiss National Science Foundation (https://www.snf.ch/en) with grants 31003A_179358 & 310030_215709 to JG. The funders had no role in study design, data collection and analysis, decision to publish, or preparation of the manuscript

**Competing interests:** The authors have declared that no competing interests exist.

Region (hereafter NCR—Chandel, 2014). While the structure and functions of the coding region tends to be well studied and remain relatively conserved, with several RNA and protein coding genes, much of the NCR's function is unknown. It has been demonstrated that the NCR region holds the promoters for the replication, with a part of this region forming a third DNA strand in a region called the D-loop, that acts as the origin for the replication of the heavy-strand [4]. However, series of tandem repeats with unknown function have also been observed in the NCR of various species.

These repetitions, different between species in terms of location, size and number, have been identified in many vertebrates including mammals—bats [5], shrews [6] and European minks [7]—and birds—yellow browed tits [8], and owls [9, 10]. Moreover, extreme variation in the number of repeats has been found at intra-individual (i.e. heteroplasmy; 6), intraspecific [8] and interspecific levels [5, 9].

Although the mechanisms by which these tandem repeats are formed is still unknown, several hypotheses including mispairing via slippage [6], Slipped-strand mispairing and insertions via self-recombination [8] have been proposed as possible mechanisms explaining this phenomenon. Moreover, and despite the presence of these repetitions in many species, few studies have, to our knowledge, tried to disentangle its relationship with phylogenetic, environmental, physiological, and/or demographic factors.

The barn owl (*Tyto alba*), a widespread raptor, offers a good opportunity to study the dynamic of these repeats with a recently published complete reference genome [11] along with many whole genome resequencing data [11–14]. Taking advantage of the new reference mitochondrial genome, we described its organization and characterized the tandem repeats of the NCR. Then, using whole genome sequences of 174 individuals from 18 western palearctic populations, we quantified the number of repeats in their respective mitochondrial genome. We finally combined this information with characteristics of each sequenced individual (such as the phylogeny based on the coding region of the mtDNA, the population of origin, the age, or the sex) and explored the possible factors associated with the variation in number of repetitions.

## Materials and methods

### Mitochondrial genome sequencing and annotation

The reference genome used in this study is described in Machado, Topaloudis, et al., 2022 [11]. Briefly, the genome was obtained from a blood sample and was sequenced through high-fidelity PacBio (Pacific Biosciences) sequencing. Long mtDNA reads were identified based on the similarity with the previous reference genome of the barn owl (NCBI Reference Sequence: NW_022670451.1) [15] and assembled into a new complete, circular genome (Accession number MZ318036.1). We annotated the genome using Mitos Web Server [16] and manually curated the annotation. Results are presented in Fig 1 and S1 Table. Tandem repeats in the reference genome were identified using tandem repeat finder online tool [17]. Based on the tandem repeats identified, the reference was trimmed to only include each of the long-repeated regions once, producing a trimmed reference genome (provided in Supplementary material).

### Whole genome re-sequencing data

A total of 172 barn owl individuals from 16 populations were used in this study: 10 from Aegean Islands (AE), 30 from Switzerland (CH), 10 from Crete (CT), 10 from Cyprus (CY), 10 from Denmark (DK), 10 from East Canary Islands, 10 from West Canary Islands, 5 from France (FR), 15 from Great Britain (GB), 10 from Greece (GR), 5 from the Ionian Islands (IO), 12 from Ireland (IR), 10 from Israel (IS), 10 from Italy (IT), 10 from Portugal (PT), 5 from Serbia (SB)

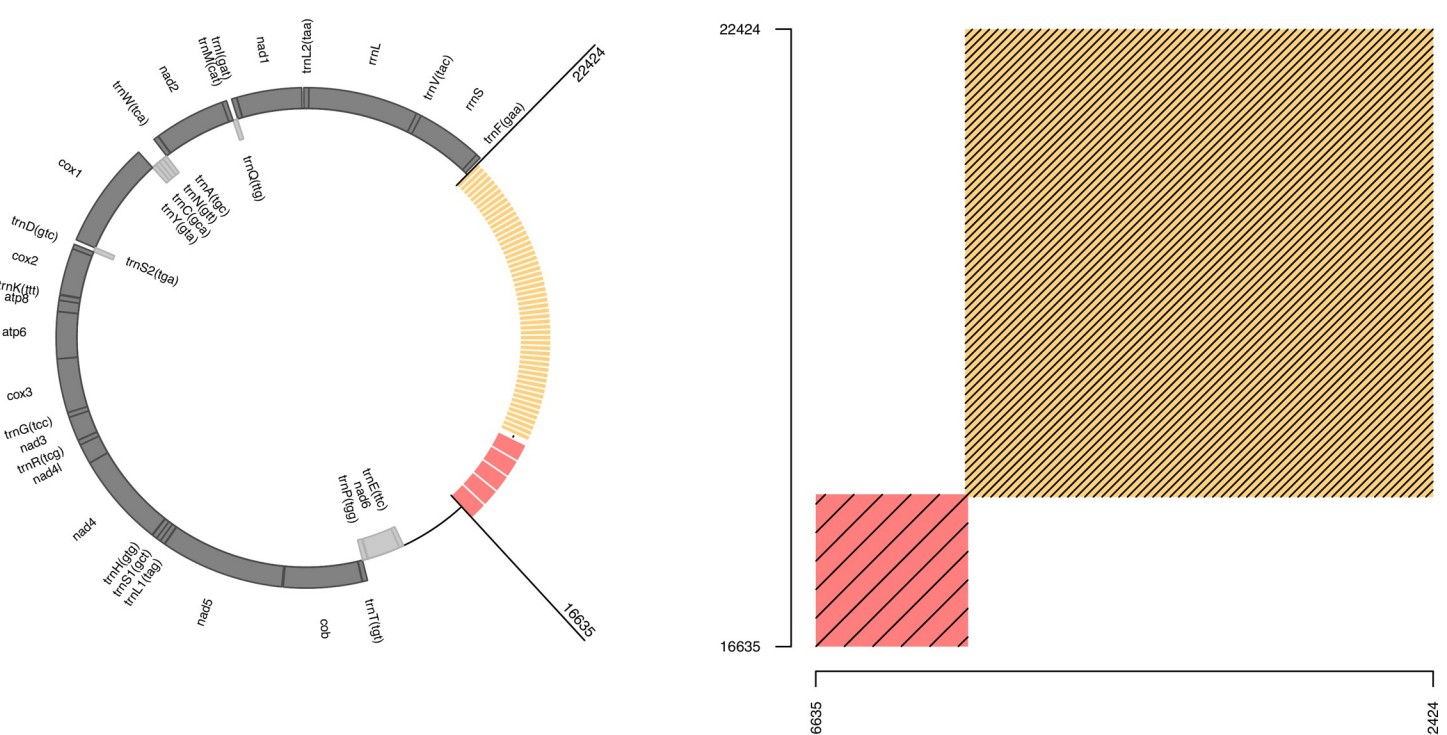

**Fig 1. The mitochondrial genome of the barn owl (*Tyto alba*).** (a) Annotation of the reference genome. Dark-grey boxes depict genes on the heavy strand. Light-grey boxes represent genes on the light strand. Coordinates of each gene can be found in the S1 Table. Red and orange boxes represent respectively the *long* and *short* tandem repeats in the Non-Coding Region. (b) Dot plot alignment of the regions harboring the repeats against itself. Each line represents two sequences with a similarity higher than 90%. *Red* and *orange* boxes highlight the *long* and *short* tandem repeats respectively.

(S1 Fig). Whole genome sequences for these individuals were published in previous studies (Cumer et al., 2022 [12], GenBank BioProject PRJNA727977; Machado, Cumer, et al., 2021 [14], BioProject PRJNA700797; Machado, Topaloudis, et al., 2022 [11], BioProject PRJNA727915, Cumer, et al., 2023 [13], BioProject PRJNA925445). Two additional individuals of other barn-owl species (one *T. furcata* from America and one *T. javanica* from Singapore; classification according to Uva et al., 2018 [18]) were included in the analyses and used as outgroups. The sequences for the outgroups were published in Machado et al., 2021 ([14]; GenBank BioProject PRJNA700797). All individuals were sequences with Illumina HiSeq 2500 high-throughput paired-end sequencing, producing 150 bp long reads. An exhaustive list of samples, their mitochondrial genome coverage as well as the sampling year, their sex, the tissue used for extraction, as well as the bioproject they are associated with is reported in table S2 Table.

## Mitochondrial genome sequence of the re-sequenced samples

Raw reads were aligned on the trimmed reference (see *Mitochondrial genome sequencing and annotation section* for details). We trimmed the reads with Trimommatic v.0.36 to remove the adapter and keep reads longer than 70bp length [19]. Trimmed reads were mapped to the trimmed reference mitochondrial genome using the BWA-MEM v.0.7.15 algorithm [20]. We then called variants using the bcftools v1.8 [21] tools mpileup (with mapping quality > 60, depth < 5000) and call (consensus calling (-c) for haploid data (ploidy = 1)). We then created a consensus fasta sequence with bcftools consensus, applying variants called above on the reference genome. Individual sequences can be downloaded at https://github.com/cumtr/Serrano_et_al_2024_Diversity_mitochondrial_genome_Barn_Owl.

## Estimation of the number of tandem repeats in the non-coding region (NCR)

To estimate the number of tandem repeats for each individual, we retrieved the sequencing depth at each position along the trimmed reference genome using Samtools v1.16 [21, 22]. The number of copies for each repeat was then computed by comparing the mean coverage of the repeated region (between 16750 and 16900 bp for the long repeat and between 17005 and 17025 bp for the short repeat) compared to the median coverage of the whole mitochondrial genome (S2 Fig). The estimated numbers of copies were then rounded to the closest integer. We also retrieved the median coverage of the coding part of mitochondrial genome (from 1 to 15542 bp) for each sample.

## Phylogenetic analysis

Based on the annotation of the reference, we identified the delimitation between the coding region and the non-coding region (at end of the trnE gene, 15542bp). After this, individual consensus sequences aligned, manually checked, and trimmed to exclude the NCR of the mitochondrial genome using Geneious Prime (version 2022.0.2). A neighbor joining tree was generated using the Ape package [23] in R version 4.2.1 [24] and PopArt [25] was used to generate a Haplotype network through the TCS method. The individuals were grouped together into haplogroups based on the genetic distances in the tree (S3 Fig) and clustering in the haplotype network (S4 Fig). Haplogroup A was further separated into 4 different subgroups (A1 to A4).

## Statistical analysis

To identify variables explaining the variation in number of copies of each of the short and long repeats, we fitted a linear model for each of the repetitive regions independently. Modeling was done using the *lm*() function in the basic stats package from R [24]. The explanatory variables used for each sequenced individual included: the sex of the individual, whether the individual was coming from an island or the mainland, the population of origin, the haplogroup, the age of the individual (encoded as juvenile or adult), as well as the tissue used for extraction. Age data was available only for some individuals (n = 83 in the reduced dataset). We also included the median coverage of the mitochondrial genome in order to control for potential bias due to non-even sequencing between samples.

The *drop1()* function from the basic *stats* package [24] was used to identify and exclude non-significant factors in order to improve the model. The resulting relevant factors were analyzed individually, combined, and taking their interactions into account through single and multi-factor ANOVAs using the *aov()* function in basic stats package [24]. An Akaike criterion test was then performed to identify the model with the best fit using *aictab()* function in *AICc-modavg* package [26]. Contingency tables were used to evaluate if the distribution of data between variables could introduce bias, and the analysis was repeated while excluding biased categories when this was the case. Finally, a Tuckey post hoc test was performed to identify differences between categories using the *TuckeyHSD()* function in the basic *stats* package [24]. Because of the non-homogeneity of the data between tissue and the age of the individuals at sampling, the analysis was repeated focusing on certain treatments: either based on the age (including only juvenile samples) or the tissue used for the extraction (using only blood samples, or only the combination of muscle or internal organ samples).

## Results

### a. Mitochondrial genome annotation

The total size of the reference genome was 22,461 bp and the NCR started at position 15,543. The annotation of the genes showed that the mitochondrial genome of the barn owl coded for 2 tRNA, 22 rRNA and 13 protein coding genes (Fig 1 and S1 Table). Annotation of repetitive sequences in the genome identified one minisatellite and two repeats of different size and period (reported in S2 Table). Hereafter, we refer to the 80bp long repeat as the *short* repeat and the 256bp long repeat as the *long* repeat.

### b. Phylogenetic analysis and haplotype network

The neighbor joining tree of the coding region shows a clear differentiation of the Western Palearctic barn owl relative to the two outgroups (Figs 2A and S2). On the same tree, we can observe that individuals do not cluster according to their population of origin, except individuals from the two Canarian populations. Haplotype network also shows a clear distinction of the two outgroups relative to the rest of the samples (S3 Fig). Based on their clustering, Western Palearctic barn owls are divided into 3 haplogroups we denominated A, B and C. Haplogroup A, the most diverse, is further subdivided into 4 haplogroups named A1 to A4. Since both population and haplogroup are not related with populations distributed across haplogroups, the two factors were considered independent for downstream analyses.

### c. Quantification of the number of repeats in re-sequenced individuals

The number of *long* repeats ranges from 4 to 38 copies in re-sequenced individuals, with a modal number of 5 repeats (Fig 2B). The number of short repeats ranges from 6 to 135 copies, with a modal number of 34 repeats (Fig 2B). The number of *short* and *long* repeats are strongly correlated ($r^2$ = 0.42, p = 2.2 e-16, S5 Fig). An ANOVA test revealed no significant relationship between the number of repeats and the haplogroup in which the individuals belong, for both *short* and *long* repeat (p > 0.1) (Fig 3).

### d. Statistical analysis

**i. *Long* repeat.** To determine which factors were related to the number copies of the *long* repeat, we generated a linear model with all the explanatory variables. Sex, haplogroup, population and the median coverage were removed as factors from the analysis since the linear model improved without them (lowest AIC; see methods for details), while tissue, age, and island (i.e. whether the individual was sampled in an island or not) were kept for further analysis. The ANOVA model that included those 3 factors and their interactions showed that tissue had the highest significance (p = 4.65 e-07), followed by the interaction between tissue and age (p = 0.00175) and the Island factor (p = 0.04). An Akaike criterion test showed that the model with the best fit for explaining the number of *long* repeats was the one that included the tissue used for extraction, the age of the individual and their interaction (AIC = 981.07, AIC Weight = 0.96). However, further contingency table analysis (S6 Fig) shows imbalance in the count of data in the different categories: island individuals are mostly adults sampled mainly from muscle and internal organs while continent individuals are mostly juveniles with blood of feather samples. The difference between age categories disappeared when we included only tissues (muscle and internal organs) from which both ages were found (S6C Fig).

When analyzing tissue as the explanatory variable on its own (p = 4.95e-07), we found that blood and feather samples were significantly different from muscle and internal organs, with the latter showing higher numbers of repeats and more variation between individuals (Fig 3).

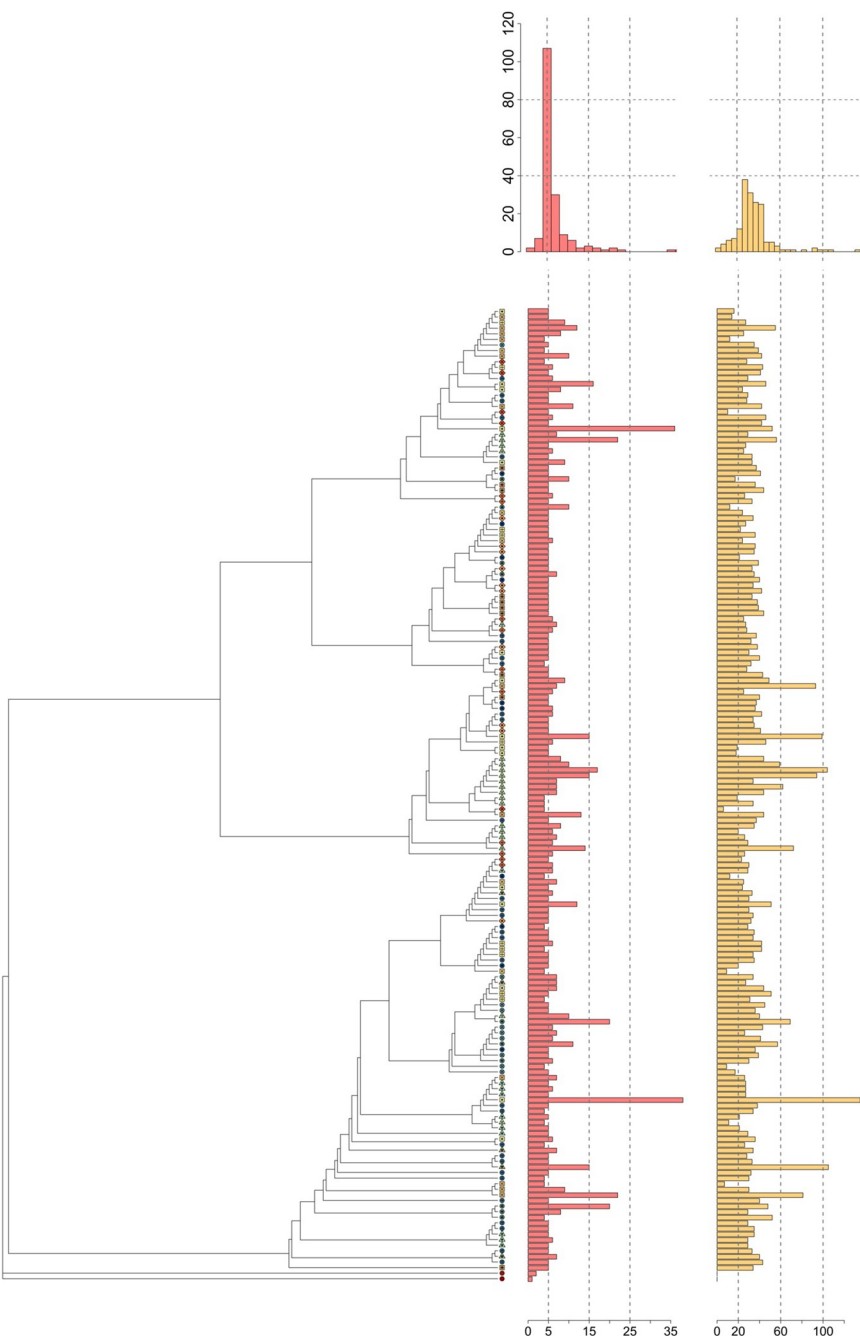

**Fig 2.** (a) Neighbor joining tree of the coding region of the mitochondrial genome of *Tyto alba*. The symbols represent each individual's population as shown in S1 Fig. The USA individual was used as the outgroup and the branch length is not represented in this figure. The original tree with branch length corresponding to genetic distances can be found as S2 Fig. For each individual in the tree, we associated (b) a bar plot of the number of repeats of both *long* (in red) and *short* (in orange) repeats in the control region. On top of each barplot, the histogram depicts the distribution of the number of each type of repeat in all the samples.

**ii.** ***Short* repeat.** In order to determine which factors were related to the number of repeats in the *short* repeat region, we fitted a linear model with all the explanatory variables. Sex, haplogroup and age were removed from the analysis as the linear model improved without them

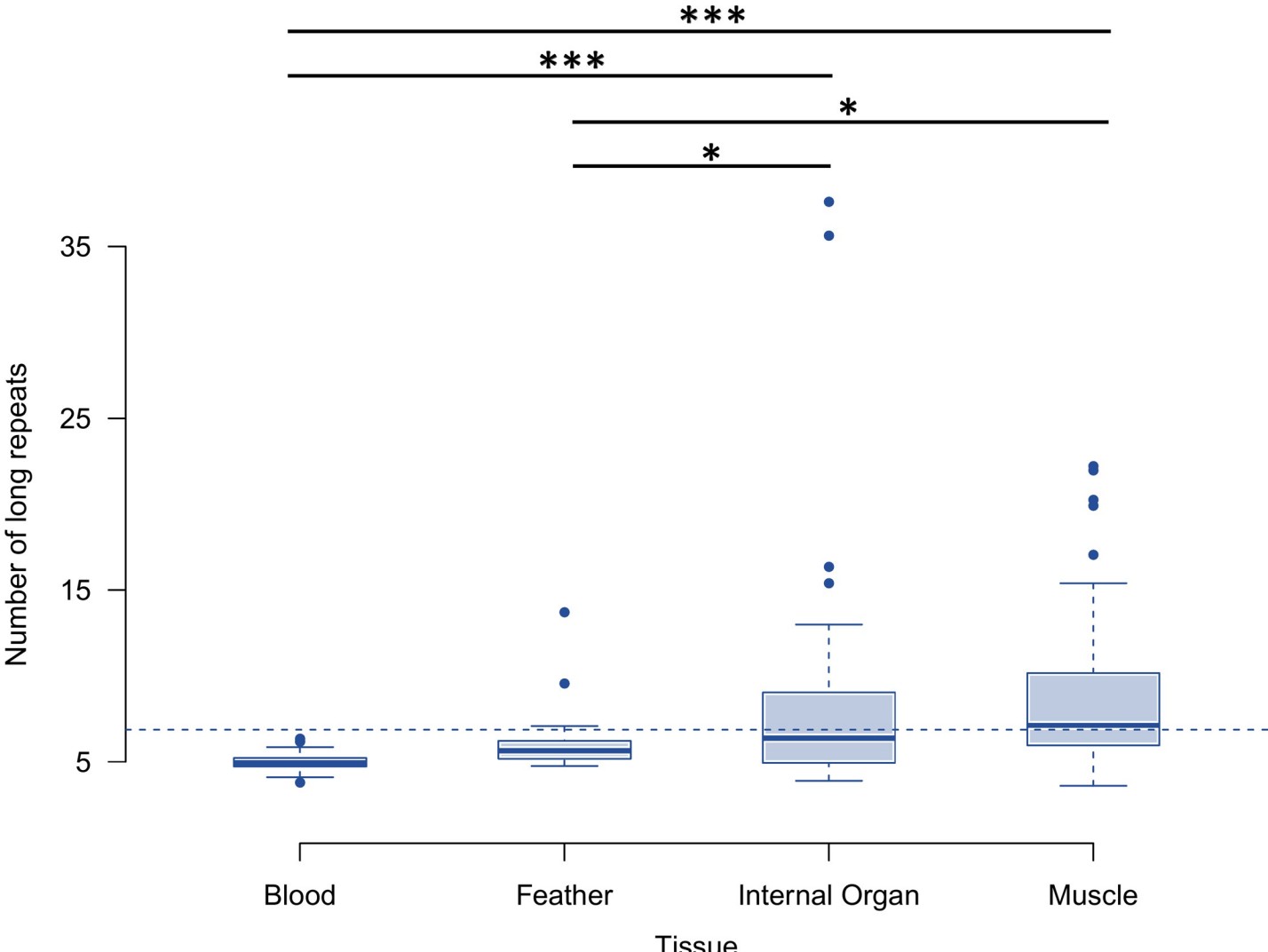

**Fig 3. Number of *long* repeats in the non-coding region of the mtDNA between individuals of the barn owl collected from different tissues.** The dashed line represents the mean number of repeats across all individuals. Bars in the top correspond to the significance of the difference between classes (From top to bottom: p = 0.00001, p = 0.00005, p = 0.045, p = 0.044), pairings with no significant difference are not shown.

(lower AIC; see methods for details). Only tissue, a nested factor between island and population and the median coverage were found to be relevant factors, with only the tissue along with the nested factor of the population within the island being significant (respective p-values: p = 0.014047 and p = 0.0004093). Contingency table showed the non-homogeneity between factors and classes, with some combinations having no individuals at all (S3 Table).

## Discussion

In this study, we described the structure of the recently produced reference mitochondrial genome of the barn owl (*Tyto alba*). Using whole genome resequencing data for more than a hundred Western Palearctic barn owls, we also quantified the variation in number of copies of two tandem repeats in the control region. Finally, we explored which factors were related to the variation in the number of repeats between individuals. All together, these results shed

light on the dynamic of variations in the number of repeats of two types of tandem repeat in the control region of the mitochondrial genome of the barn owl.

This study is the first study describing and quantifying the tandem repeats in the mitochondrial genome of the barn owl. This was made possible because of the use of whole genome sequencing with high fidelity long reads and shows that, despite a golden age of mitochondrial DNA study in the 90[ths], this organelle genome still has secrets to reveal. Indeed, the complete sequence of the mitochondrial genome of the barn owl allows to describe the organization of the coding regions and will permit comparison of its organization with other owls' genomes, thus informing about its evolution. Interestingly, this new assembly revealed two classes of repeats in the Control Region (CR), hereafter denominated *long* (256bp) and *short* (80bp) repeats, for which we investigated the dynamic using re-sequencing data for several western palearctic barn owls.

We then characterized the variation of the number of repeats between individuals. First, it is important to note that the implementation of a PCR free sequencing method allows us to discard the possibility of the number of repeats being influenced by the amplification. Second, we notice huge variations in the number of copies between individuals for both the *long* and the *short* repeat (Fig 2). Interestingly, the NCR for the two outgroups (i.e. USA and SGP, at the bottom of the tree in Fig 2A) show no amplification of the repeats identified in the reference genome. Regarding the profiles of the alignments of the re-sequencing reads for those individuals (S7 Fig), we can observe that the profile of coverage is different for those two samples. The divergence of these samples with the individual used as a reference might explain this profile. Indeed, these two samples come from different species (*Tyto furcata* and *Tyto javanica* for the USA and SGP samples respectively) while all other samples belong to the same species (*Tyto alba*), rendering the mapping for these two samples harder. However, this difference might as well be a real biological signal, with the amplifications restricted to the *Tyto alba* clade. A reference genome for both the American (*Tyto furcata*) and the Australasian (*Tyto javanica*) Owl, and/or more resequencing data for both clades will allow to better understand the evolutionary origin of these repeats.

Within the *Tyto alba* samples, we explored the factors influencing the number of *long* and *short* repeats. Some important caveats should be considered when interpreting the results. The individuals were sampled over a large time span and in different locations, resulting in different sampling procedures between them. Due to this non homogeneity of the sampling procedure, the samples cover different ages or sexes. This heterogeneity is also observable in the tissues sampled, as well as the in the ways samples were conserved (i.e. time between sampling and freezing). We thus observed a sampling bias toward some combination of the evaluated factors (i.e. adults were samples only from muscles and internal organ). This imbalance of the data limits our ability to disentangle the effect of factors such as tissue, age, or sex. Indeed, we initially observed a difference between adult and juvenile individuals (S6A Fig). However, the contingency tables showed that none of the adults were sampled from blood or feather while most of the juveniles came from these tissues (S6B Fig). The difference between age categories disappeared when we included only tissues from which both ages were found (S6C Fig). Moreover, this subsampling left few individuals to be analyzed and therefore prevents us from properly testing this factor. A similar situation was found between the island/mainland factor and the tissue used for extraction.

If the effect of tissue on the number of *short* and *long* repeats is confirmed, it could be due to the differences in the mitochondrial usage of the different tissues. A study published in 2019 found that the number of mtDNA copies was constant between tissues [27], suggesting the variation in number of repeats is not caused by differences in the replication rate. However, that same study found that tissues with a higher OXPHOS activity were subject to higher levels

of oxidative stress, therefore have a higher prevalence of mutations like insertions or deletions and higher self-recombination occurrences. Similarly, a study on close to 80,000 somatic mutations in mice mtDNA found that their composition and accumulation rates varied between different tissues [28]. These findings are consistent with our results when considering that muscle and some internal organs present high levels of OXPHOS activity [29] and that recombination has been proposed as the most probable formation mechanism for the tandem repeats in yellow browed tits [8]. The same study suggested that the secondary structure formed by the repeats may play a role in the mechanism of the formation. For this reason, it would be highly informative to study the secondary structure of the repeats in the Barn Owl, explore their stability and compare it to what has been found in other species.

It has also been proposed that the accumulation of tandem repeats may be influenced by the age of the individuals [5, 30]; The presence of different numbers of repetitions could have an effect on mitochondrial function and that the accumulation of such repeats may contribute to aging. This phenomenon may also simply be caused by older individuals having a larger time to accumulate variations [5]. We did not find evidence for this hypothesis in our study. This is consistent with some recent studies in which the presence of two mtDNA classes is suggested. Respiring or "working" molecules perform OXOHOS activity, while "replicating" molecules are responsible for renewing the mtDNA pool. This way, replicating molecules are protected from oxidative stress and avoid the accumulation of mutations with age [31]. However, as discussed earlier, the low number of adult individuals makes it difficult to properly evaluate this hypothesis. Additionally, it is known that there are differences in the renewal rate of cells between tissues [32]. Non-dividing tissues like brain cells and some internal organs present a higher presence of senescent cells [33]. In such cells, variation in the tandem repeats may still have had more chances for formation and accumulation. Therefore, another interesting question is whether the variation in repetitions depends on the age of the cells rather than the individuals.

In the future, it will be informative to track the same individuals across time and sample different tissues (such as blood or feather). This would allow to describe the variation of the repetition within different tissues of the same individuals through time. It would also be important to determine whether the variation in the tandem repeats affect mitochondrial function and health. For humans, it has been found that mutations in the NCR of the mtDNA might be related to certain cancers and other diseases [3, 4]. Finally, if the number of copies does increase with the age of individuals, this factor may explain the origin of strong correlation between number of *short* and *long* repeats (S5 Fig), a hypothesis that deserve further investigations.

## Supporting information

**S1 Table. Annotation of the reference mitochondrial genome of Tyto alba according to Mitos Web Server (Bernt et al., 2013).** Start and stop refer to the base pair positions for each gene. The + symbol represents the heavy strand and the–the light strand of the mtDNA.
(XLSX)

**S2 Table. Description of the repeats identified in the reference genome of the barn owl (Tyto alba).**
(CSV)

**S3 Table. Description of the samples used in this study.**
(CSV)

**S4 Table. Contingency table between the nested factor (island and population) and tissue used for the extraction of mtDNA of the barn owl (Tyto alba).** The numbers inside the table represent the number of individuals.
(XLSX)

**S1 Fig. Geographical location and population of the re-sequenced individuals of Tyto alba.** A. Individuals collected in Europe and its surroundings. B. Sampling locations including out-groups from USA and Singapore. Population symbols correspond to the ones shown in Fig 2.
(TIF)

**S2 Fig. Sequencing depth for one re-sequenced individual (ID: IC17) of Tyto alba aligned to the trimmed reference (with only one repeat of each repeated region).** (A) Depth along the complete genome. (B) Close up on the non-coding region where the repeats are located. The vertical lines represent the delimitation of the plateau where the mean depth for the repeated regions. The horizontal straight line represents the median depth for all base pairs in the two panels.
(TIF)

**S3 Fig. Neighbor joining tree of the coding region of the mitochondrial genome of Tyto alba resequenced individuals.** Branch lengths correspond to the genetic distance between individuals.
(TIF)

**S4 Fig. Haplotype network of the coding region of the mitochondrial genome of the re-sequenced Barn Owls (Tyto alba) used in this study.** Different colors represent the hap-logroups in which the individuals were clustered. The individuals that didn't cluster clearly with any group were excluded from the analysis. USA and Singapore individuals were also discarded based on their high differentiation from the rest of the sample. Each tick on the network branches represents a mutation.
(TIF)

**S5 Fig. Correlation between the number of repeats in both regions of the control region of the mtDNA of the resequenced individuals of Tyto alba and the difference between tissues used for the extraction.** Convex hulls were drawn to show the distribution of number of repeats between the different tissues. There is a strong correlation between both numbers of repeats $r = 0.65$, $r^2 = 0.42$, $p = 2x10^{-16}$.
(TIF)

**S6 Fig. Analysis of age as an explanatory factor for the number of long repeats in the mtDNA of Tyto alba.** (A) The number of repeats using age as the explanatory variable. (B) Heatmap of the contingency table of age in regard to tissue. Higher number of individuals corresponds to a higher color intensity. (C) The number of repeats using age as the explanatory factor while using only individuals that were collected form muscle or internal organs.
(TIF)

**S7 Fig. Sequencing depth for re-sequenced individuals from USA (*T. furcata*) and Singa-pore (*T. javanica*) aligned to the trimmed reference (with only one repeat of each repeated region).** (A) Depth along the complete genome. (B) Close up on the non-coding region where the repeats are located in *T. alba*. The horizontal straight line represents the median depth for all base pairs in the two panels.
(TIF)

## Author Contributions

**Conceptualization:** Jérôme Goudet, Tristan Cumer.

**Data curation:** Tristan Cumer.

**Formal analysis:** Marcelo J. Serrano, Tristan Cumer.

**Funding acquisition:** Jérôme Goudet.

**Methodology:** Marcelo J. Serrano, Jérôme Goudet, Tristan Cumer.

**Supervision:** Tristan Cumer.

**Writing – original draft:** Marcelo J. Serrano, Tristan Cumer.

**Writing – review & editing:** Jérôme Goudet, Tristan Cumer.

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
