## [Decision Letter · Decision Letter 0]

2 Aug 2023

PONE-D-23-20889Characterization of the diversity of barn owl’s mitochondrial genome reveals high copy number variations in the control region.PLOS ONE

Dear Dr. Cumer,

Thank you for submitting your manuscript to PLOS ONE. After careful consideration, we feel that it has merit but does not fully meet PLOS ONE’s publication criteria as it currently stands. Therefore, we invite you to submit a revised version of the manuscript that addresses the points raised during the review process.

We look forward to receiving your revised manuscript.

Kind regards,

Sven Winter

Academic Editor

PLOS ONE

Journal Requirements:

   "This work was supported by the Swiss National Science Foundation (https://www.snf.ch/en) with grants 31003A_179358 & 310030_215709 to JG."

5. We note that supplementary Figure 1 in your submission contain map/satellite images which may be copyrighted. All PLOS content is published under the Creative Commons Attribution License (CC BY 4.0), which means that the manuscript, images, and Supporting Information files will be freely available online, and any third party is permitted to access, download, copy, distribute, and use these materials in any way, even commercially, with proper attribution. For these reasons, we cannot publish previously copyrighted maps or satellite images created using proprietary data, such as Google software (Google Maps, Street View, and Earth). For more information, see our copyright guidelines: http://journals.plos.org/plosone/s/licenses-and-copyright.

a. You may seek permission from the original copyright holder of supplementary Figure 1 to publish the content specifically under the CC BY 4.0 license.  

Additional Editor Comments:

One of the reviewers raised important concerns about study design and the drawn conclusions of the manuscript and I fully agree with these concerns. Therefore, I would suggest that the study design and questions studies are carefully reconsidered and the necessary revisions made before resubmission.

Reviewers' comments:

Reviewer's Responses to Questions

**Comments to the Author**

1. Is the manuscript technically sound, and do the data support the conclusions?

Reviewer #1: Yes

Reviewer #2: Partly

2. Has the statistical analysis been performed appropriately and rigorously? 

Reviewer #1: Yes

Reviewer #2: Yes

3. Have the authors made all data underlying the findings in their manuscript fully available?

Reviewer #1: Yes

Reviewer #2: Yes

4. Is the manuscript presented in an intelligible fashion and written in standard English?

Reviewer #1: Yes

Reviewer #2: Yes

5. Review Comments to the Author

Reviewer #1: I think this article would be suitable for publication. This article could be useful for conservation policy of endangered owls later on. I hope the authors need to continue more investigation of endangered owl for the future.

Reviewer #2: The manuscript submitted by Serrano et al. describes substantial length variation observed in the control region of Tyto alba. The authors used a previously mitochondrial genome they obtained (Machado et al. 2022, MZ318036.1) that they here annotated. On genbank, the mtdna genome was specified to have rather low coverage (10X). The annotation provided by the authors would suggest very unusual features (in face unique to bird to my knowledge and i have checked or sequenced litterally hundred of complete mt dna genomes), which includes 5 independent duplication of five coding loci (ND2, CO1, ATP8, ND3, ND5) ! I then checked the Suppl. Table 1 and realised that it is not duplication but more likely wrong automatic annotation (it is actually a bit embarassing for the authors to not have checked this). The authors may perform manual curation of the annotation and compare it with closely related species for start end end of loci ; for example ND2, if merging _a and _b would be 1047 bp which is plausible (usually it is 1041 bp but a 2 codon insertion is possible).

Remaining analyses appear to be well done but i have a few questions regarding sequencing biases and how it would impact conclusions regarding the impact of age/island, etc

I actually wonder if there could not be across individual variation in the number of sequenced reads and thus inference in the number of repeats ? Could not we figure that the sequencing (hence coverage) of repeated regions can be more variable (with higher variance) than less repeated regions ? That tissue type would be linked to sequencing biases rather than number of copies per se ? What about numts numts ? How is overall coverage (feather versus muscle versus blood) ? For organ samples, could there be a bias linked to sample preservation ? for example, blood or feather will be preserved immediatly whereas it is likely that for organs there would be time between death and freeze or death and sampling and potentially different freeze/thaw cycles and thus sample degradation. All of the hypotheses tested by the authors are very interesting but i wonder what could actually be said with. I think the authors have something very interesting to go after but as they noted, they can not properly test most factors so i am unsure what could be concluded. For this purpose a careful sampled design may be needed to forrmely test their hypotheses.

Minor comments :

‘widespread’ may be more correct than ‘widely spread’

6. PLOS authors have the option to publish the peer review history of their article (what does this mean?). If published, this will include your full peer review and any attached files.

Reviewer #1: No

Reviewer #2: No

---

## [Author Response · Author response to Decision Letter 0]

13 Sep 2023

PONE-D-23-20889

Characterization of the diversity of barn owl’s mitochondrial genome reveals high copy number variations in the control region.

PLOS ONE

Journal Requirements:

[R] We have now formatted our manuscript according to style requirements.

[R] We removed this sentence in the new version of the manuscript.

 "This work was supported by the Swiss National Science Foundation (https://www.snf.ch/en) with grants 31003A_179358 & 310030_215709 to JG."

[R] We confirm that: "The funders had no role in study design, data collection and analysis, decision to publish, or preparation of the manuscript."

[R] The new version of the manuscript includes the caption of the Supporting Information.

5. We note that supplementary Figure 1 in your submission contain map/satellite images which may be copyrighted. All PLOS content is published under the Creative Commons Attribution License (CC BY 4.0), which means that the manuscript, images, and Supporting Information files will be freely available online, and any third party is permitted to access, download, copy, distribute, and use these materials in any way, even commercially, with proper attribution. For these reasons, we cannot publish previously copyrighted maps or satellite images created using proprietary data, such as Google software (Google Maps, Street View, and Earth). For more information, see our copyright guidelines: http://journals.plos.org/plosone/s/licenses-and-copyright.

a. You may seek permission from the original copyright holder of supplementary Figure 1 to publish the content specifically under the CC BY 4.0 license. 

[R] We have changed the supplementary Figure 1. It now uses the maps from Natural Earth (public domain) and should be suitable for publication.

Additional Editor Comments

One of the reviewers raised important concerns about study design and the drawn conclusions of the manuscript and I fully agree with these concerns. Therefore, I would suggest that the study design and questions studies are carefully reconsidered and the necessary revisions made before resubmission.

[R] We have addressed all the comments raided by the reviewers. Please find bellow our point-by-point answers.

Review Comments to the Author

Reviewer #1: I think this article would be suitable for publication. This article could be useful for conservation policy of endangered owls later on. I hope the authors need to continue more investigation of endangered owl for the future.

[R] We are thankful of the referee for his/her positive feedbacks on our work.

Reviewer #2: The manuscript submitted by Serrano et al. describes substantial length variation observed in the control region of Tyto alba. The authors used a previously mitochondrial genome they obtained (Machado et al. 2022, MZ318036.1) that they here annotated. On genbank, the mtdna genome was specified to have rather low coverage (10X). The annotation provided by the authors would suggest very unusual features (in face unique to bird to my knowledge and i have checked or sequenced litterally hundred of complete mt dna genomes), which includes 5 independent duplication of five coding loci (ND2, CO1, ATP8, ND3, ND5) ! I then checked the Suppl. Table 1 and realised that it is not duplication but more likely wrong automatic annotation (it is actually a bit embarassing for the authors to not have checked this). The authors may perform manual curation of the annotation and compare it with closely related species for start end end of loci ; for example ND2, if merging _a and _b would be 1047 bp which is plausible (usually it is 1041 bp but a 2 codon insertion is possible).

[R] We thank the reviewer for pointing to this issue with the automatic annotation. As we mentioned in our previous version of the manuscript, we did not made the differentiation between spited annotation and duplicated genes (“Split or duplicated genes included […]”).

We have now manually curated the annotation, and all the genes mentioned were indeed spited in the automatic annotation. We have now changed the annotation and modified the main text and figure 1 accordingly. See figure 1 and line 88:

“We annotated the genome using Mitos Web Server (16) and manually curated the annotation.”

We have also removed the sentence mentioned above about spited and duplicated genes, see line 179.

Remaining analyses appear to be well done but i have a few questions regarding sequencing biases and how it would impact conclusions regarding the impact of age/island, etc

I actually wonder if there could not be across individual variation in the number of sequenced reads and thus inference in the number of repeats ? Could not we figure that the sequencing (hence coverage) of repeated regions can be more variable (with higher variance) than less repeated regions ? That tissue type would be linked to sequencing biases rather than number of copies per se ? What about numts numts ? How is overall coverage (feather versus muscle versus blood) ? For organ samples, could there be a bias linked to sample preservation ? for example, blood or feather will be preserved immediatly whereas it is likely that for organs there would be time between death and freeze or death and sampling and potentially different freeze/thaw cycles and thus sample degradation. All of the hypotheses tested by the authors are very interesting but i wonder what could actually be said with. I think the authors have something very interesting to go after but as they noted, they can not properly test most factors so i am unsure what could be concluded. For this purpose a careful sampled design may be needed to forrmely test their hypotheses.

[R] We agree with the reviewer that multiple factors can bias our results.

Regarding the tissue used, this is something we mentioned in our previous version. We have now added a caution note about the way tissue were conserved. see line 306: 

“The individuals were sampled over a large time span and in different locations, resulting in different sampling procedures between them. Due to this non homogeneity of the sampling procedure, the samples cover different ages or sexes. This heterogeneity is also observable in the tissues sampled, as well as the in the ways samples were conserved (i.e. time between sampling and freezing). We thus observed a sampling bias toward some combination of the evaluated factors (i.e. adults were samples only from muscles and internal organ). This imbalance of the data limits our ability to disentangle the effect of factors such as tissue, age or sex.”

Sequencing biases might also have consequences on our results, but we think our approach managed to overcome most of the biases.

All the individuals were sequenced to have a genome-wide coverage of around 15X (see Machado et al. 2021, Cumer et al. 2022, Cumer et al. 2023 and Machado et al. 2022 for details). However, we do observe inter-individual variation in the number of sequenced reads attributed to the mitochondrial genome, with a median individual coverage varying between 35 X and 24269 X, and a median of 606 X. Also, we did observe a link between the overall coverage and the tissue used for extraction, with higher coverage in muscles compared to other tissues (see attached figure). We think this might be due to the higher number of mitochondria in muscles, hence a higher proportion of mtDNA compared to genomic DNA.

We could also observe a positive relation between the median coverage of the coding region and the coverage of the repeated region. This is expected and shows that the repeated reads come from the repeated region of the mitochondrial genome.

As explained line 127, “The number of copies for each repeat was then computed by comparing the mean coverage of the repeated region (between 16750 and 16900 bp for the long repeat and between 17005 and 17025 bp for the short repeat) compared to the median coverage of the whole mitochondrial genome (Sup. Fig. 2).” After this normalization, the number of short and long repeats negatively correlate with the median coverage, meaning that less covered mitochondrial genomes seems to have more repeats. 

To account for the potential effect of the total coverage of the mitochondrial genome, we re-ran the models including the median coverage of the coding region. The new version of the manuscript extensively describes the results. In brief, including this factor in our models did not change the results. For the long repeat, the median coverage was excluded from the model since models without this factor had a better fit. For the short repeat, the median coverage was kept in the final model but did not have a significant contribution (p-value = 0.17).

Regarding the nuclear mitochondrial DNA (numt), no numts were ever reported in the barn owl genome and a BLAST of the mitochondrial genome on the reference genome confirmed this result (data not sown). We cannot exclude the presence of unassembled numts in the genome, however, with a mean genomic coverage of 15X, the contribution of a hypothetic numt to our estimation of number of copies in the mitochondrial genome seems marginal.

Finally, we agree with the reviewer that a carefully sampled design may be needed to formally test our hypotheses, something we mention line 349: 

“In the future, it will be informative to track the same individuals across time and sample different tissues (such as blood or feathers). This would allow to describe the variation of the repetition within different tissues of the same individuals through time”

Minor comments :

‘widespread’ may be more correct than ‘widely spread’

[R] changed, see line 69.

---

## [Decision Letter · Decision Letter 1]

3 Nov 2023

PONE-D-23-20889R1Characterization of the diversity of barn owl’s mitochondrial genome reveals high copy number variations in the control region.PLOS ONE

Dear Dr. Cumer,

Thank you for submitting your manuscript to PLOS ONE. After careful consideration, we feel that it has merit but does not fully meet PLOS ONE’s publication criteria as it currently stands. Therefore, we invite you to submit a revised version of the manuscript that addresses the points raised during the review process.

We look forward to receiving your revised manuscript.

Kind regards,

Sven Winter

Academic Editor

PLOS ONE

Journal Requirements:

**Additional Editor Comments:**

The revised manuscript did greatly improve and the reviewer, who has seen the revision, seems satisfied with the authors replies. I agree with the reviewer that all generated data (rawdata and assembled data) needs to be submitted to a public database before the manuscript can be accepted. Please also ensure, that all figures are submitted in a high enough resolution for publishing. 

Reviewers' comments:

Reviewer's Responses to Questions

**Comments to the Author**

1. If the authors have adequately addressed your comments raised in a previous round of review and you feel that this manuscript is now acceptable for publication, you may indicate that here to bypass the “Comments to the Author” section, enter your conflict of interest statement in the “Confidential to Editor” section, and submit your "Accept" recommendation.

Reviewer #2: All comments have been addressed

2. Is the manuscript technically sound, and do the data support the conclusions?

Reviewer #2: Yes

3. Has the statistical analysis been performed appropriately and rigorously? 

Reviewer #2: Yes

4. Have the authors made all data underlying the findings in their manuscript fully available?

Reviewer #2: No

5. Is the manuscript presented in an intelligible fashion and written in standard English?

Reviewer #2: Yes

6. Review Comments to the Author

Reviewer #2: The authors may need to submit all the assembled mitochondrial genomes to Genbank or EMBL to make the data avalaible to researchers interesting in replicating the analyses performed in the analyses. Figure is not available in the revised version but the version in the first version was very blurry.

Apart from this all comments that could be addressed were addressed properly

7. PLOS authors have the option to publish the peer review history of their article (what does this mean?). If published, this will include your full peer review and any attached files.

Reviewer #2: No

---

## [Author Response · Author response to Decision Letter 1]

24 Nov 2023

Journal Requirements:

Additional Editor Comments:

The revised manuscript did greatly improve and the reviewer, who has seen the revision, seems satisfied with the authors replies. I agree with the reviewer that all generated data (rawdata and assembled data) needs to be submitted to a public database before the manuscript can be accepted. Please also ensure, that all figures are submitted in a high enough resolution for publishing. 

[R] We have now deposited all the data used for this publication on public databases:

- Raw reads can be found via the NCBI website. The bioprojects for the different individuals can be found in table S2.

- The mitochondrial genome for each individuals, the table of the coverage along the MT genome used to estimate the number of copies in each individual genome, the alignment used to build the tree in Figure 2, as well as the reference genome used in this study can now be found here :

https://github.com/cumtr/Serrano_et_al_2023_Diversity_mitochondrial_genome_Barn_Owl

We have now specified this information line 122 of the manuscript:

“Individual sequences can be downloaded at https://github.com/cumtr/Serrano_et_al_2023_Diversity_mitochondrial_genome_Barn_Owl”

Reviewers' comments:

Reviewer's Responses to Questions

Comments to the Author

1. If the authors have adequately addressed your comments raised in a previous round of review and you feel that this manuscript is now acceptable for publication, you may indicate that here to bypass the “Comments to the Author” section, enter your conflict of interest statement in the “Confidential to Editor” section, and submit your "Accept" recommendation.

Reviewer #2: All comments have been addressed

2. Is the manuscript technically sound, and do the data support the conclusions?

Reviewer #2: Yes

3. Has the statistical analysis been performed appropriately and rigorously? 

Reviewer #2: Yes

4. Have the authors made all data underlying the findings in their manuscript fully available?

Reviewer #2: No

[R] All the data used in this paper are now available online. See our comment above for details.

5. Is the manuscript presented in an intelligible fashion and written in standard English?

Reviewer #2: Yes

6. Review Comments to the Author

Reviewer #2: The authors may need to submit all the assembled mitochondrial genomes to Genbank or EMBL to make the data avalaible to researchers interesting in replicating the analyses performed in the analyses. Figure is not available in the revised version but the version in the first version was very blurry.

Apart from this all comments that could be addressed were addressed properly

[R] All the data used in this paper are now available online. See our comment above for details.

We have also uploaded figures in high quality for publication.

---

## [Editor Report · Decision Letter 2]

27 Nov 2023

Characterization of the diversity of barn owl’s mitochondrial genome reveals high copy number variations in the control region.

PONE-D-23-20889R2

Dear Dr. Cumer,

We’re pleased to inform you that your manuscript has been judged scientifically suitable for publication and will be formally accepted for publication once it meets all outstanding technical requirements.

Kind regards,

Sven Winter

Academic Editor

PLOS ONE
---

## [Editor Report · Acceptance letter]

1 Dec 2023

PONE-D-23-20889R2 

Characterization of the diversity of barn owl’s mitochondrial genome reveals high copy number variations in the control region. 

Dear Dr. Cumer:

I'm pleased to inform you that your manuscript has been deemed suitable for publication in PLOS ONE. Congratulations! Your manuscript is now with our production department. 

Kind regards, 

on behalf of

Dr. Sven Winter 

Academic Editor

PLOS ONE